# Traumatic Brain Injury Leads to Alterations in Contusional Cortical miRNAs Involved in Dementia

**DOI:** 10.3390/biom12101457

**Published:** 2022-10-11

**Authors:** Shahmir Naseer, Laura Abelleira-Hervas, Dhwani Savani, Ross de Burgh, Robertas Aleksynas, Cornelius K. Donat, Nelofer Syed, Magdalena Sastre

**Affiliations:** Department of Brain Sciences, Imperial College London, Hammersmith Hospital, Du Cane Road, London W12 0NN, UK

**Keywords:** traumatic brain injury, Alzheimer’s disease, neuroinflammation, miRNAs, BACE1, GSK3-β

## Abstract

There is compelling evidence that head injury is a significant environmental risk factor for Alzheimer’s disease (AD) and that a history of traumatic brain injury (TBI) accelerates the onset of AD. Amyloid-β plaques and tau aggregates have been observed in the post-mortem brains of TBI patients; however, the mechanisms leading to AD neuropathology in TBI are still unknown. In this study, we hypothesized that focal TBI induces changes in miRNA expression in and around affected areas, resulting in the altered expression of genes involved in neurodegeneration and AD pathology. For this purpose, we performed a miRNA array in extracts from rats subjected to experimental TBI, using the controlled cortical impact (CCI) model. In and around the contusion, we observed alterations of miRNAs associated with dementia/AD, compared to the contralateral side. Specifically, the expression of miR-9 was significantly upregulated, while miR-29b, miR-34a, miR-106b, miR-181a and miR-107 were downregulated. Via qPCR, we confirmed these results in an additional group of injured rats when compared to naïve animals. Interestingly, the changes in those miRNAs were concomitant with alterations in the gene expression of mRNAs involved in amyloid generation and tau pathology, such as β-APP cleaving enzyme (BACE1) and Glycogen synthase-3-β (GSK3β). In addition increased levels of neuroinflammatory markers (TNF-α), glial activation, neuronal loss, and tau phosphorylation were observed in pericontusional areas. Therefore, our results suggest that the secondary injury cascade in TBI affects miRNAs regulating the expression of genes involved in AD dementia.

## 1. Introduction

Dementia currently affects around 50 million people worldwide. The aetiology of Alzheimer disease (AD) remains unclear, but it is likely to involve both genetic and environmental factors. Serious head injury is a significant environmental risk factor for the subsequent development of AD. A history of traumatic brain injury (TBI) accelerates the onset of AD [1,2] and the more severe the injury, the greater the risk of developing AD [3]. Worldwide, it has been estimated that TBI affects over 54 to 60 million people annually, leading to either hospitalisation or mortality. Of all types of injury, those to the brain are among the most likely to result in death or permanent disability. There were about 61,000 TBI-related deaths in the United States in 2019 (Center for Disease Control and Prevention). In the UK, the National Institute of Clinical Excellence (NICE) estimates that each year, 1.4 million people visit Accident & Emergency departments with head injuries, of which 200,000 are admitted for treatment. Therefore, it is imperative to investigate the mechanisms linking head injury and the development of dementia later in life.

The presence of amyloid-β (Aβ) plaques in TBI patients when examining brain tissue surgically removed from patients or post-mortem brains has suggested a link between TBI and AD [4,5,6]. However, recent in vivo imaging studies have shown that the plaque distribution differs from that observed in AD patients [7]. There are indications that a long-term process of Aβ metabolism is initiated by TBI through an increase in the expression of the enzymes involved in Aβ generation, such as BACE1 [8,9], as well as of the amyloid precursor protein (APP) [4,10,11]. In addition, neurofibrillary tangles (NFT) have been observed in TBI brains, particularly in chronic traumatic encephalopathy (CTE) cases. Other aetiologies in common between TBI and AD include blood–brain barrier breakdown, neuronal loss, axonal damage, glial activation, the release of free oxygen radicals, and acetylcholine deficiency [12,13,14,15].

Early after TBI, in both patients and animal models, the activation of microglia is evident throughout the injury site [16]. Remarkably, the activation of microglial cells continues for years after the initial injury and can be detected in both in vivo and post-mortem [16,17,18]. Due to the fact that microglial activation occurs at the same site as the neuronal degeneration and axonal damage following TBI [19,20], it has been hypothesized that this inflammatory response provides a mechanistic link between TBI and AD. Therefore, the understanding of the relationship between TBI and AD neurodegeneration is of relevance in order to find new therapeutic targets [16,21] and biomarkers for dementia.

In this study, we aimed to explore pathways that might link TBI with AD, with the idea that TBI could induce aberrant epigenetic regulation of processes critical for normal brain function and thereby affecting the progression to AD. Epigenetic modifications, such as histone modification and DNA methylation, affect electrostatic repulsion between histone proteins, altering the chromatin condensation and the access to the transcriptional machinery [22]. Others involve anomalies in the regulatory actions of small nucleolar RNAs and microRNAs. There is evidence of alterations in epigenetic markers in TBI, such as increased levels of histone-3 acetylation and DNA hypomethylation or alterations in micro-RNAs, such as miR-23a and miR-27a [23,24,25,26]. While these studies highlight global epigenetic changes after TBI, they did not provide any molecular insights by which epigenetic mechanisms link TBI with the pathological hallmarks of dementia. Interestingly, plasma miRNAs profiling in human TBI have revealed the dynamic temporal regulation of miRNA expression within the cortex, suggesting that they could be used as potential biomarkers [27].

Here, we investigated whether miRNAs that have been involved in neurological disorders, particularly AD, are altered in the brain of a moderate/severe model of TBI two weeks post-injury. Our hypothesis is that TBI induces changes in specific miRNA expression particularly involved in neurodegeneration, neuroinflammation, and the development of Aβ plaques and neurofibrillary tangles, since acute systemic inflammation secondary to TBI could impact epigenetic modifications [28].

## 2. Materials and Methods

### 2.1. Animals

For this study, we used a controlled cortical impact (CCI) model of TBI in adult male wild-type Sprague Dawley rats, of approximately 2 months of age (around 250 g), which were obtained from PsychoGenics. Control naïve rats from Charles River were used and kept in appropriate conditions regarding food, water, and light/dark cycles. Rats had ad-libitum access to standard rodent chow (Rat and Mouse No.1 Maintenance, Special diets service, UK). The CCI was created with a cortical contusion device. Rats were anaesthetized with 5% isoflurane and oxygen, mounted in a stereotaxic frame, the scalp was retracted, and a drill was used to expose the brain. CCI was induced with a 5-mm-diameter rounded brass impactor attached to a computer-controlled piston (Custom Design & Fabrication, Virginia Commonwealth University, Richmond, VA, USA) propelled electronically with following parameters: velocity = 2.5 m/s; depth = 3 mm; duration = 100 ms). Rats were then given appropriate post-operative care and left for 2 weeks. For tissue extraction, all animals were subjected to deep pentobarbital (Euthanal, Boehringer Ingelheim, Bracknell, UK) anaesthesia. After cessation of all pain reflexes, they were perfused via the transcardial route using ice-cold PBS (0.01 M phosphate buffer, 0.0027 M potassium chloride, and 0.137 M sodium chloride, pH 7.4), using the Leica Perfusion One system at approximately 50 bar and a 20 g blunt needle. The tissue was frozen in isopentane cooled with liquid nitrogen and kept at −80 °C. During the course of the study, 12/12 light/dark cycles were maintained. The room temperature was be maintained between 20 °C and 23 °C with a relative humidity maintained around 50%. Chow and water were provided ad libitum for the duration of the study.

For the immunohistochemistry experiments, sections were obtained from our previous study [29], using a similar severity (bevelled-flat impactor, velocity = 4 m/s; depth = 2 mm; duration = 100 ms, Leica Impact One). Naïve animals were only subjected to terminal pentobarbital anaesthesia and perfused with PBS.

All studies involving animals were reported in accordance with the ARRIVE guidelines. The experimental procedures were performed in accordance with the guidelines reported in EU Directive 2010/63/EU.

### 2.2. mRNA Extraction

For the mRNA and miRNA extractions, mirVana miRNA isolation kit (Thermofisher, Oxford, UK) was used. Part of the ipsilateral cortex containing the injury and the correspondent contralateral tissue from the TBI rat brain, as well as the same brain region of naive control rats were homogenised using a minilys personal homogenizer (Bertin instruments, Steventon, UK) containing lysis/binding solution. The lysed tissue/cells were incubated on ice for 10 min in 1/10 of miRNA homogenate additive. After the incubation, 300 μL Acid-Phenol:Chloroform was added (equal to the initial lysate volume), mixed for 30 s, and centrifuged for 5 min at maximum speed (10,000 rpm). Carefully, the aqueous phase was transferred to a fresh Eppendorf tube. For the enrichment of small RNAs, 100% ethanol was added to the aqueous solution and mixed thoroughly. Next, the samples were passed through a filter cartridge and the filtrates were collected after the centrifugation at 10,000 rpm for 15 s. At this stage, the filtrate contains the small miRNAs while the filter still has total mRNAs. So, the filters were kept for the total mRNA enrichment. The filtrates were mixed with ethanol and passed through a new filtrate. The recovered RNAs were stored at −20 °C.

### 2.3. miRNA Array

We carried out a miRNA array (miScript miRNA PCR Array, Qiagen, Manchester, UK) to detect and quantify the levels of miRNAs associated with rat neurological and development disease. For the RT PCR reaction, 200 ng of extracted miRNA was added to MiScript II RT Kit (Qiagen) for the generation of cDNA from the small RNA extracted. Following this, the array was performed using SYBR green reagent kit (miScript SYBR Green PCR Kit, Qiagen), comparing miRNA extracts from the area of the injury with the contralateral area of the cortex.

### 2.4. RT-qPCR

For miRNA analysis, the miScript SYBR green PCR kit (Qiagen) was used following the suppliers quick-start protocol. The primers for miR-9, miR-29b, miR-34a, miR-106b, miR-181a, and miR-107, as well as the house-keeping control Hs-SNORD61_11, were obtained from Qiagen.

For the mRNA RT-qPCR, quatifast SYBR PCR Kit (Qiagen) was used according to the manufacturers protocol and the genes were normalized to GAPDH (see primers Table 1).

### 2.5. Immunohistochemistry

Immunohistochemistry (IHC) was conducted to analyse the localization, density, and morphology of microglia, astrocytes and neurons using IBA-1, GFAP and NeuN antibodies, respectively. In addition, we performed staining for BACE1 and phosphorylated tau (*p*-tau). Paraffin brain sections were deparaffinized in xylene and hydrated using decreasing concentrations of ethanol (100%, 100%, 90%, 70%, distilled H_2_O). Sections were washed in 1X PBS and submerged in 1% H_2_O_2_ in PBS-Triton-X 0.3% for 30 min for tissue permeabilization and the quenching of endogenous peroxidase activity. Sections were incubated in heated 0.01 M citrate buffer (pH 6) for 20 min in a steam bath for antigen retrieval. Slides were cooled on ice and rinsed in 1X PBS. Sections were then incubated overnight with primary antibodies Iba-1 (Wako) 1:5000; GFAP (DAKO), 1:5000, NeuN (Millipore) 1:2000, and p-tau CP13 (from Peter Davies) at 1:200) diluted in PBS-Triton-X 0.3% at 4 °C in a humidified chamber. Super Sensitive™ Polymer-HRP IHC detection system (BioGenex, Fremont, CA, USA) was used to detect the primary antibody. DAB activity was quickly quenched with Milli-Q and tap water and sections were counterstained with Mayer’s Haematoxylin (TCS Biosciences, Buckingham, UK) for 5 min, developed under running tap water, and dehydrated via increasing concentrations of ethanol (70%, 90%, 100%, 100%) and xylene. Finally, coverslips were mounted using DPX mounting solution (Thermo Scientific, Oxford, UK).

### 2.6. Image Analysis

Samples were imaged with Axio Imager slide scanner using a 10X objective. Quantification of staining was performed with HALO software (Indica Labs, UK). Selected areas were quantified (see schematic atlas in immunohistochemistry figures) in at least 3 independent sections for each animal. Optimized microglia module with set analysis parameters was used to measure cell number (per mm^2^) and average process area (reported in μm) [29].

### 2.7. Statistics

Analysis was performed with GraphPad Prism 8 (v8.02). RT-qPCR data were analysed using double-delta CT method. Obtained data were compared using one-way or two-way ANOVA analysis followed by post hoc (Tukey’s multiple comparisons test) analysis for multiple groups and Student’s *t*-test for two groups. Values were considered statistically significant when *p* < 0.05. Data are presented as mean ± standard error of the mean (SEM).

## 3. Results

### 3.1. miRNAs Related to Dementia Are Altered in the CCI Brain

To determine whether miRNAs related to dementia or neurodegeneration were altered in TBI, we conducted a miRNA array assay (miScript miRNA PCR Array, Qiagen). This kit contains mature miRNA-specific forward primers (miScript Primer Assays) that have been arrayed in biologically relevant pathway-focused (in this case for neurological development and disease), in miRNA extracts from the CCI rat model of TBI. We compared miRNA extracts from the pericontusional cortical area of the injury with the corresponding contralateral area of the cortex. In the injured cortex, substantial alterations in the levels of certain miRNAs associated with dementia/AD (Figure 1A) were observed. Due to the large amount of data, only those miRNAs that showed a fold change larger than +/− 0.2 were included in the graphs. Data gathered from the miRNA array for both ipsilateral and contralateral hemispheres were calculated using the ΔΔCT method.

Among the miRNAs with relevant changes between both areas, we detected the decreased levels of miR-29b, which had previously been linked with the regulation of β-APP cleaving enzyme-1 (BACE1) levels [30] and increases in miR-9, which was implicated in microglial activation [31] (Figure 1A). In addition, miR-106b was found to be decreased in the injured cortex, which has been involved in the inhibition of tau phosphorylation [32].

To confirm these results, we then performed qPCR for a selection of miRNAs that were related to AD pathology [33] and that had shown a fold change between 0.2–1.7 when comparing the ipsilateral with the contralateral side. qPCR was performed in cortical extracts from the ipsilateral and contralateral sides of the same CCI rats and compared to the values of those obtained from naïve rats as controls (Figure 1B–G).

miR-29b1, miR-34a_1, miR-106b_1, miR-181a_2, and miR-107_2 were significantly downregulated in the brains of injured animals when compared to naïve rats (miR-29b_1 *p* = 0.0042; miR-34a_1 *p* = 0.0161; miR-106b_1 *p* = 0.0037; miR-181a_2 *p* = 0.0006; miR-107_2 *p* = 0.0001) (Figure 1C–G), while miR-9_1 was found upregulated in CCI rat brains (*p* = 0.0177) (Figure 1B). The results of this analysis confirmed and complemented the results from the miRNA microarray, suggesting that indeed there is a dysregulation in miRNAs involved in AD pathology and neuroinflammation in TBI brains.

### 3.2. TBI Affects the mRNA Levels of BACE1 and GSK3β

We found that some of the miRNAs altered in the TBI brains have been involved in the regulation of mRNA levels of genes that are known to be involved in AD (Figure 1). Following this, the mRNA analysis of genes, such as *APP, BACE1*, *GSK3**β*, and *MAPT* was conducted in the same brains.

Because a downregulation in the levels of miR-29b_1 and miR-107_2 was detected in TBI brains, which are involved in the regulation of *BACE1* mRNA, we determined the expression level of *BACE1* mRNA. Our results show that the mRNA expression of *BACE1* was significantly upregulated by 63% (*p*-value, 0.0345) in the injured cortex when compared to that of naïve animals (Figure 2A). Interestingly, an upregulation of the BACE1 protein level was previously reported in rat brains following CCI [8]. We did not observe any changes in the contralateral side.

Because Glycogen synthase-3-β (GSK3β) contributes to the tau hyperphosphorylation and Aβ generation [34,35,36] in the pathogenesis of AD, we therefore investigated the mRNA expression levels of GSK3β in the brains of CCI-injured and compared them to that of naïve rats. GSK3β mRNA was significantly upregulated by 62% (*p*-value, 0.0423) at the ipsilateral side of the injured brain compared with naïve control rats (Figure 2B). There was also an increase in the expression of GSK3β contralateral to the injury but it was not significant when compared to the naïve brains (*p*-value, 0.0575).

Additionally, we determined the levels of other genes involved in AD and TBI, such as APP and MAPT, but did not detect any differences between TBI and naïve brains (Figure 2C,D).

### 3.3. CCI Induces Changes in miRNAs Involved in the Inflammatory Response

Because our TBI model displayed changes in miR-9 and miR-181, reported to be involved in the regulation of inflammatory genes as well as microglia and astrocyte activation, the mRNA levels of inflammatory markers, including TNF-α, IL-6, Fractalkine, and IL-4, were analysed in CCI and naïve rats.

The expression level of TNF-α (*Tnf*) mRNA was significantly increased by 405% at the ipsilateral side of TBI brains compared to naïve rats (Figure 2E, *p*-value, 0.0007) and the contralateral side of the injury (*p* = 0.0005).

The mRNA expression level of IL-6 (*Il-6*) was also upregulated in TBI rat brains (271%) compared to naïve rats; however, the results did not reach statistical significance (Figure 2F).

In contrast to IL-6 and TNFα increases, the expression of the anti-inflammatory cytokine IL-4 (*Il-4*) was downregulated by 50% in the TBI rat brains compared with sham control and the contralateral side of the injury; however, those changes were not statistically significant (Figure 2G).

The mRNA expression of fractalkine (*Fractalkine*) was also downregulated by 56% at the ipsilateral side and 50% at contralateral side of the injury when compared with naïve control brains; however, those changes did not reach statistical significance (*p*-values = 0.08 and 0.12) (Figure 2H). Using a different type of statistical comparison, unpaired t-tests showed a significant reduction between the ipsilateral side of the TBI brain and the naïve brain (*p* = 0.02). No changes in fractalkine expression were observed between the ipsilateral and contralateral of animals subjected to CCI.

### 3.4. Effect of TBI on the Density of Glial Cells and Neurons in the Brain

Microglia and astrocytes are activated following TBI, resulting in the secretion of cytokines and chemokines [16]. To understand the phenotype of glial cells in this particular model of CCI at 2 weeks post-injury, we performed immunohistochemistry with Iba-1 and GFAP and determined changes in distribution, density, and morphology.

The sections of naïve animals exhibited ramified microglia throughout all cortical layers, corresponding to a non-activated, resting state (Figure 3B). Ramified microglia were identified by their small cell bodies with long, fine processes. In addition, the contralateral cortex of the CCI rats also showed similar ramified microglia and their number was not significantly increased following injury compared with naïve animals (Figure 3B,E). In contrast, microglia on the ipsilateral side of the injury, especially around the pericontusional border, displayed a more reactive phenotype. This is demonstrated by enlarged soma with short, thick processes or amoeboid morphology with large cell bodies and no processes (Figure 3B,C). There was also a massive and significant increase in IBA-1-positive cells in the ipsilateral cortex by day 15 compared with the contralateral cortex (*p* < 0.0001, n = 5) (Figure 3E), as well as the ipsilateral (*p* < 0.0001, n = 4) and contralateral (*p* < 0.0001, n = 4) cortices of naïve animals (Figure 3E). This reactive phenotype was substantiated by changes in the area covered by IBA-1 positive processes. Our analysis revealed that the average process area in the ipsilateral cortex of injured animals was significantly reduced when compared to the cortex of naïve (*p* < 0.0001, n = 4) animals and to the contralateral side of the CCI rats (*p* < 0.05), confirming the retraction of processes (Figure 3F).

In contrast, ramified microglia in the hippocampus were evenly distributed in naïve and CCI animals and there was no statistically significant change in the average process area per microglia when compared with naïve animals (Figure 3H). Similar to the cortex, there was a significant upregulation of microglia density in the ipsilateral hippocampus at 15 days post-injury (dpi) compared to the ipsilateral hippocampus of naïve rats (*p* < 0.0001, n = 4) (Figure 3G). The contralateral hippocampus of 15 dpi animals also exhibited a significant increase in IBA-1-positive cells when compared to the contralateral hippocampus of naïve animals (*p* < 0.0001, n = 4) (Figure 3G).

Reactive astrocytosis was examined by GFAP staining. In all naïve rats, quiescent astrocytes were present throughout all six cortical layers, lined the ependymal layer and pial surface, and connected to blood vessels (Figure 4B). Quiescent astrocytes were recognized by their small cell bodies and long, fine processes (Figure 4B). Following TBI, an upregulation of reactive astrocytes, characterized by enlarged cell bodies with thick, short processes, was notable in the ipsilateral cortex of injured animals (Figure 4B,C). GFAP reactivity was more evident in the pericontusional cortex, forming an astrocytic scar (Figure 4B). The density of GFAP-positive cells was significantly increased in the ipsilateral cortex 15 dpi when compared to the contralateral cortex (*p <* 0.0001, n = 5) and with the cortices of naïve animals (Figure 4E) (*p* < 0.0001, n = 4–5). The percentage area covered by GFAP was significantly increased in the ipsilateral side of the injury compared with the contralateral side and with naïve controls (Figure 4F) (*p* < 0.0001, n = 4–5).

Although increased GFAP staining was evident in the hippocampus 15 dpi compared with naïve animals, no statistical significance was reached (Figure 4G). In addition, no differences were found when comparing the area covered by GFAP+ astrocytes in the hippocampus among different groups (Figure 4H).

### 3.5. Characterization of Neuronal Loss in Cortex of Hippocampus of CCI Rats

We next determined the consequences of TBI on neuronal loss 2 weeks post-injury, whereby neuronal death occurs either as a consequence of the mechanical forces of the injury or as result of secondary processes. Immunohistochemical staining using the neuron-specific nuclear protein NeuN revealed a characteristic pattern of healthy neurons showing round or oval morphology, while necrotic or apoptotic neurons within the injured tissue appeared small and irregularly shaped (Figure 5B).

In all injured animals, NeuN staining decreased dramatically in the ipsilateral cortex, particularly in the impact region, when compared to sham/naïve animals (Figure 5B,E). The neuron count in the ipsilateral cortex of injured rats was significantly decreased as compared to the contralateral cortex (*p* < 0.0001, n = 7), ipsilateral (*p* < 0.0001, n = 4–7), and contralateral (*p* < 0.0001, n = 4–7) cortices of sham/naïve animals (Figure 5E).

However, no neuronal loss was observed in the hippocampus of CCI rats. NeuN staining in the ipsilateral CA1 and CA3 hippocampal regions of TBI animals was no notable different compared to sham/naïve animals or the contralateral side of the injury (Figure 5F,G).

### 3.6. Alterations in the Expression of Proteins Involved in Alzheimer’s Disease Pathology in the CCI Model

We next assessed the protein levels of genes involved in Aβ generation and tau phosphorylation in the CCI model. Because the mRNA levels of BACE1 and GSK3, which is a kinase involved in tau phosphorylation, were found upregulated in CCI brains, we carried out immunohistochemical analysis to determine the protein expression and distribution of BACE1 and phosphorylated tau (*p*-tau) in the CCI model, 2 weeks post-injury. We did not observe differences in BACE1 expression levels 2 weeks post-injury (data not shown).

The effect of moderate TBI on the phosphorylation of tau protein was analysed by immunostaining with CP13 antibody, which specifically detects tau proteins with phosphorylation epitope around phospho-serine 202. An increase in CP13 staining was observed in the ipsilateral side of the cortex of the CCI rats compared with naïve (*p* = 0.0002) and the contralateral side (*p* = 0.0026) (Figure 6A,B,E). No significant changes were detected in the hippocampus (Figure 6C,F).

## 4. Discussion

The aim of this study was to investigate the link between TBI and AD by examining the alterations in the expression of certain miRNAs involved in the regulation of AD-related genes in a CCI model. Indeed, our results revealed that two weeks after a head injury, miR-29b, miR-107b, miR-34, and miR-181 were significantly downregulated following CCI, compared to naïve rat brains, while miR-9 was found to be upregulated post-injury. We speculate that the alterations in these miRNAs could be secondary to the inflammatory response that occurs after injury, since it has been documented that epigenetic processes instruct gene-specific reprogramming during acute systemic inflammation [28]. Indeed, alterations in the expression of these miRNAs were concomitant with changes in the expression of inflammatory markers (cytokines, chemokines, and glial alterations), neuronal death and in the expression of genes involved in Aβ generation and tau phosphorylation (such as BACE1 and GSK3).

Although there are numerous reports investigating the alterations of miRNAs following TBI, these studies were mostly performed in blood or CSF as potential biomarkers [26,37]. The studies carried out in brains of animal models of TBI have revealed different patterns, depending on the time post-injury, the brain region investigated, the model used, and the severity of the injury [23,38,39,40,41]. For instance, one of the first reports carrying out miRNA studies in animal models of TBI reported altered expression levels of miR-107, -130a, -223, -292-5p, -433-3p, -451, -541, and -711 immediately post-injury [42]. On the other hand, 7 days post-injury, seven miRNAs (miR-144, miR-136, miR-148b-5p, miR-135a, miR-135b, miR-342-5p, and miR-190) were found to be reduced, while three miRNAs (miR-23a, miR-363, and miR-130b) were increased [38]. Studies performed 2 weeks post-injury are scarce [43] and do not involve the analysis of the pericontusional tissue. A recent meta-analysis of 34 studies performed in severe TBI models revealed that the most common pathways regulated after TBI by miRNAs involve the TNFα and endocytosis signalling [41]. Interestingly, the list of miRNAs involved included miR-107 and miR-181b, which were also found altered in our study.

Similarly, a clear association of alterations in certain miRNAs has been reported for AD, both in patients and in animal models. In particular, Cogswell and colleagues showed the dysregulation of around 300 miRNA regionally and stage-specific in human AD brain [44], which were involved in amyloid processing, neurogenesis, and innate immunity. Additionally, a microarray analysis has revealed that almost 60 miRs were differentially expressed in AD mouse brain [45].

The significant downregulation of miR-29b by TBI shown in our investigation has important implications in AD, since this miRNA family seems to be involved in the regulation of APP and BACE1 expression [30]. Interestingly, the levels of this miRNA have also been found reduced in AD patients and in animal models of AD [30,46]. In particular, Hebert and colleagues found that miR-29 is a direct target for BACE1 in its 3′ UTR region and showed that the overexpression of miR-29a/b-1 in HEK-293 cells severely decreased the expression of BACE1 and the secretion of Aβ [30]. Another miRNA that regulates BACE1 expression is miR-107 [47,48]. miR-107 was found downregulated in temporal cortex of AD brains [30] and in the brains of CCI rats in the present study. The consequent upregulation of BACE1 mRNA observed in our TBI model, probably a consequence of reduced miR-107 and miR-29b levels, did not coincide with changes at the protein level, which is in line with previous studies showing increases in BACE1 mRNA and proteins, only detectable shortly after brain injury [8,49]. These results are in agreement with our previous studies showing that BACE1 expression is upregulated by inflammatory cytokines [50].

Other potential signalling mechanisms regulating APP and BACE1 expression involve GSK3-β activity [34,35], which is also implicated in tau phosphorylation [36]. Importantly, our study shows an upregulation of the GSK3β gene expression in the CCI model. GSK3 is a putative target of numerous miRs identified in neural tissue, such as miR-132, miR-539-3p, and indirectly miR-21 [51]. Interestingly, we observed an increase in tau phosphorylation in pericontusional areas. These results suggest that miRNA-GSK3β regulation could have potential therapeutic relevance in TBI and there are various preclinical studies showing the beneficial effects of GSK3β inhibition in TBI models [52].

miR-29b has also been associated with cell survival. Work by Kole and colleagues in 2011 illustrated that miR-29 can act as an inhibitor of neuronal apoptosis [53]. Ideally, most neurons are expected to survive for the entire life span of an organism as they are post-mitotic and therefore have limited capability of regeneration, unlike other cells of the body. In AD and in TBI, neuronal loss is evident at later stages and it was detected in our model in the pericontusional cortex but not in the hippocampus. However, it is believed that the low expression of miR-29 could be linked to a high level of apoptotic markers in the AD brain [54,55]. In fact, the brain-specific knockdown of miR-29 led to cell death in large areas of hippocampus and cerebellum, confirming the important role of miR-29 in neuronal cell survival [56].

In the present work, we found a significant upregulation of miR-9 in the CCI model, also reported in previous studies [39,57,58]. miR–9 is highly expressed in the central nervous system and has been reported to be altered in AD brains [33,59]. The upregulation of miR-9 has been previously linked with microglia activation by targeting monocyte chemotactic protein-induced protein 1 (MCPIP1) and the downstream activation of NF-κB [60]. MCPIP1 was reported to control inflammation by negatively regulating macrophage activation, acting as an anti-inflammatory mediator [61]. In addition, miR-9 plays an important role in the differentiation of neural stem cells [62,63]. Furthermore, miR-9 promotes the proliferation of neuronal progenitors by targeting stathimin, which is crucial for the regulation of the cell cytoskeleton; therefore, miR-9 increases microtubule instability by targeting stathimin [64]. Furthermore, miR-9 regulates or controls axonal extension and branching through targeting the levels of Map1b in mouse cortical neurons, which is also critical in the stability of microtubule in the cytoskeleton [65]. Interestingly, the upregulation of miR-9 promoted synaptic remodelling in Alzheimer’s disease [66].

We also observed a significant reduction in the levels of miR-181a in CCI brains compared to naïve rats. Previous studies have pointed out the important role played by miR-181 in regulating TNF-α-induced apoptosis in both in vivo and in vitro [67] and the activation of microglia and astrocytes in CNS diseases [68] The downregulation of miR-181 has also been shown in acute ischemic stroke patients [69]. In addition, miR-181a has been reported to promote anti-inflammatory effects by decreasing the levels of inflammatory mediators and reactive oxygen species in macrophages and monocytes [70].

miR-34a, which showed a significant downregulation in our TBI model, has been linked to the regulation of tau protein expression and to repress the expression of endogenous tau proteins in human neuroblastoma cell lines M17D [71]. It is believed that miR-34a targets the 3’UTR region of tau and lowers its expression. The increased expression of tau protein in TBI rat brains could potentially be modulated by the low expression of miR-34a, although our study did not show differences in MAPT gene expression in the CCI model. In addition, miR-34a directly inhibits Bcl2 and XIAP, both anti-apoptotic proteins [72]; therefore, low levels of miR-34a could lead to higher neuronal death.

Another miRNA that targets tau is miR-106b, which inhibits tau phosphorylation at Tyr18 by targeting Fyn in a model of AD [32]. miR-106b has been reported to be downregulated in temporal cortex of AD brain and is involved in the regulation of ABCA1 levels [59].

Together, this study suggests a clear association between miRNAs altered post-injury and the expression of genes involved in the development of dementia. Therefore, these results are key to understanding the link between TBI and dementia. Notably, the role of miRNAs could be essential to understand the molecular mechanisms leading to secondary brain damage, including the regulation of genes involved in glial activation, neuronal regeneration, and the apoptosis and disruption of the BBB [37].

In addition, these data could have translational implications, including the use of miRNAs as potential biomarkers [27,73] or as therapeutic targets by using miRNAs mimics (agomirs) and miRNA inhibitors (antagomirs) [58,74]. Therefore, our results provide grounds for potential new targets for therapeutic intervention in TBI and AD and contribute to the understanding of the association between both disorders.

## Figures and Tables

**Figure 1 biomolecules-12-01457-f001:**
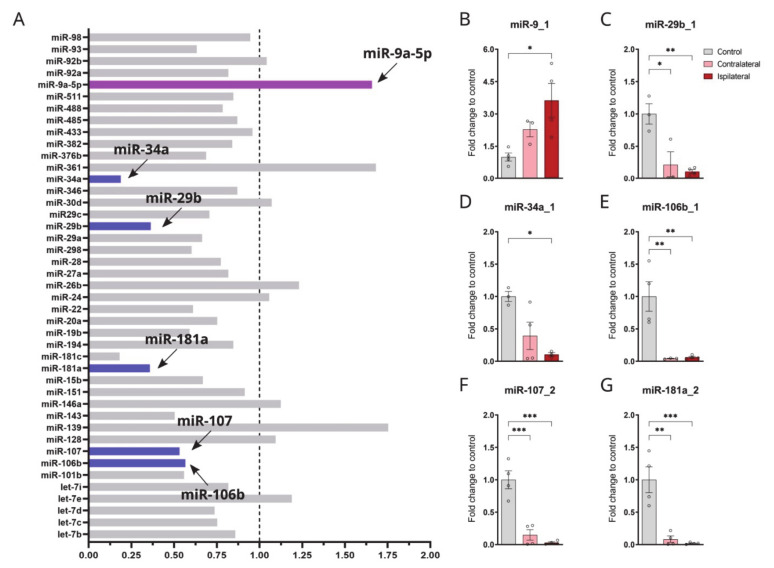
TBI alters the expression of miRNAs involved in neurological disorders. (**A**) Results obtained from miScript miRNA PCR array focused on miRNAs involved in neurological development and disease, comparing samples obtained in the ipsilateral hemisphere with the contralateral hemisphere in a rat model of CCI. Due to the large cohort of data, only those miRNAs that showed a fold change larger than +/− 0.2 were included in the graph. (**B**–**G**) Quantification of miRNA expression analysis by qPCR from rat CCI brain (ipsilateral and contralateral sides) and naïve rats (n = 4 per group). (**B**) quantification of miR-9_1 (one-way ANOVA F = 6.390 *p* = 0.0220); (**C**) miR-29_1 (one-way ANOVA F = 13.54 *p* = 0.0039); (**D**) miR-34a_1 (one-way ANOVA F = 7.614 *p* = 0.0175); (**E**) miR-106b_1 (one-way ANOVA F = 15.07 *p* = 0.0019); (**F**) miR-107_2 (one-way ANOVA F = 32.48 *p* < 0.0001); (**G**) miR-181a_2 (one-way ANOVA F = 21.50 *p* = 0.0004). Columns represent mean ± SEM, One-way ANOVA (Tukey’s multiple comparisons test) * *p* < 0.05, ** *p* < 0.01 and *** *p* < 0.001.

**Figure 2 biomolecules-12-01457-f002:**
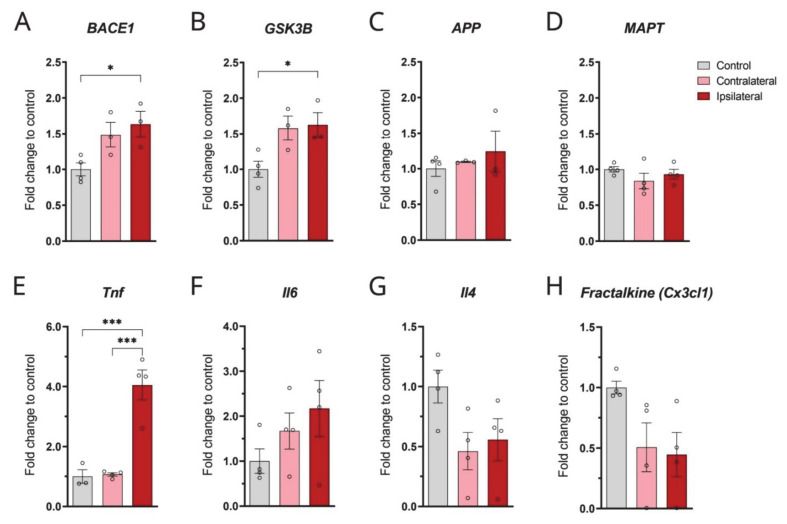
mRNAs levels of genes associated with AD and neuroinflammation are dysregulated in the cortex of the CCI rat model. mRNA expression analysis was carried out in rat CCI brains, comparing the ipsilateral and contralateral sides with naïve rat brains (n = 4 per group). (**A**) Quantification of *BACE1* (one-way ANOVA F = 5.901 *p* = 0.0315); (**B**) *GSK3**β* (one-way ANOVA F = 6.138 *p* = 0.0289); (**C**) *APP* (one-way ANOVA F = 0.5521 *p* = 0.5989); (**D**) *MAPT* (one-way ANOVA F = 1.137 *p* = 0.3629); (**E**) *Tnf* (one-way ANOVA F = 27.65 *p* = 0.0003)*;* (**F**) *Il-6* (one-way ANOVA F = 1.648 *p* = 0.2455); (**G**) *Il-4* (one-way ANOVA F = 3.366 *p* = 0.0810); (**H**) *Cx3cl1* (Fractalkine) (one-way ANOVA F = 3.620 *p* = 0.0702) mRNAs. Values shown in graphs represent the mean value ± SEM and are expressed as fold change of naïve control rats. Statistical analysis included one-way ANOVA (Tukey’s multiple comparisons test) * *p* < 0.05, and *** *p* < 0.001.

**Figure 3 biomolecules-12-01457-f003:**
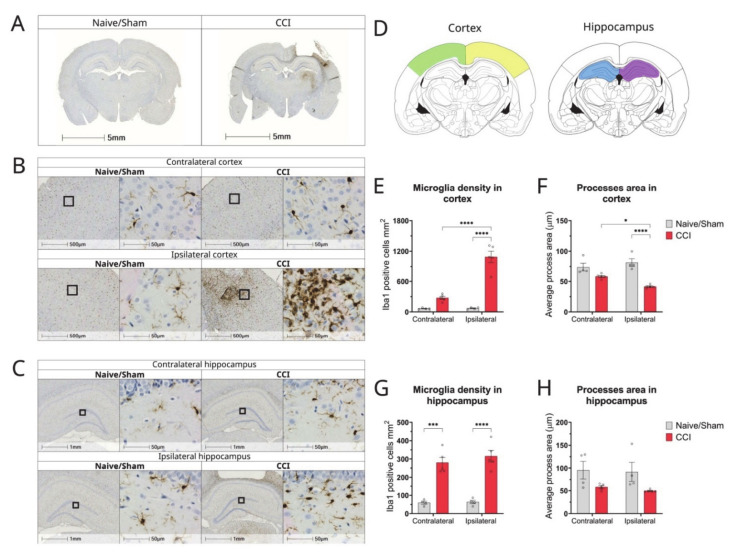
Changes in microglial density, morphology, and distribution in cortex and hippocampus of CCI rat brains 2 weeks post-injury. (**A**) Representative whole brain photomicrographs of Iba1 + staining in naïve/control and CCI rat brains; (**B**) Magnified representative images of the cortex of naïve/sham animals and CCI brains from ipsilateral and contralateral sides of the injury; (**C**) Magnified representative images of the hippocampus of naïve/sham animals and CCI brains from ipsilateral and contralateral sides of the injury; (**D**) Brain atlas images, showing areas of cortex and hippocampus analysed; (**E**) Quantification of the microglia count/mm^2^ by Iba1 staining in the cortex of CCI and sham/naïve control rats (2-way ANOVA hemisphere F(1,14) = 38.65 *p* < 0.0001, injury F(1,14) = 86.36 *p* < 0.0001, interaction F(1,14) = 37.63 *p* < 0.0001); (**F**) Average process area per microglia in cortex of CCI and naïve/sham rats (2-way ANOVA hemisphere F(1,14) = 1.129 *p* = 0.3060, injury F(1,14) = 44.53 *p* < 0.0001, interaction F(1,14) = 8.569 *p* = 0.0110); (**G**) Quantification of the microglia count/mm^2^ in the hippocampus of CCI and sham/naïve control rats (2-way ANOVA hemisphere F(1,14) = 0.631 *p* = 0.4402, injury F(1,14) = 9 3.32 *p* < 0.0001, interaction F(1,14) = 0.409 *p* = 0.5324); (**H**) Average process area per microglia in hippocampus of CCI and naïve/sham rats (2-way ANOVA hemisphere F(1,14) = 0.234 *p* = 0.6357, injury F(1,14) = 9.955 *p* = 0.0070, interaction F(1,14) = 0.036 *p* = 0.8522);. All data are presented as mean ± SEM, n = 4–5 per group. Two-way ANOVA, * *p* < 0.05, *** *p* < 0.001; **** *p* < 0.0001. Mouse brain atlas images were obtained from the Allen Institute website (www.alleninstitute.org, accessed on 29 September 2022).

**Figure 4 biomolecules-12-01457-f004:**
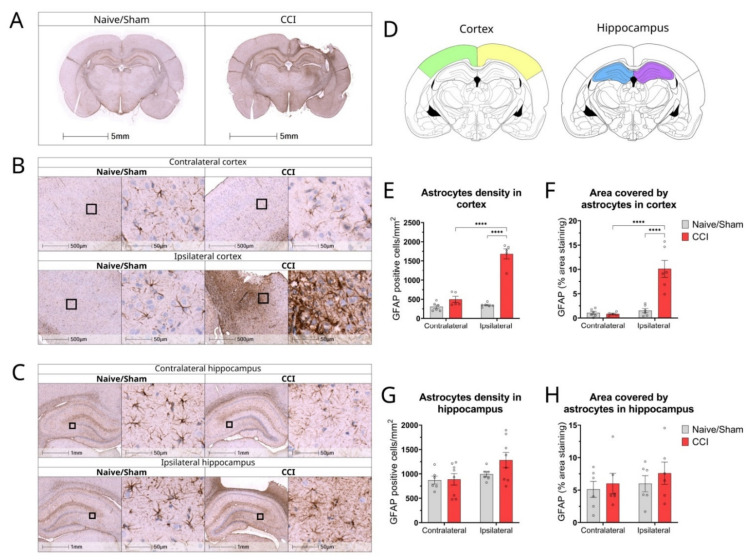
Changes in astrocyte density, morphology and distribution in cortex and hippocampus of CCI rat brains 2 weeks post-injury. (**A**) Representative whole brain photomicrographs of GFAP+ staining in naïve/control and CCI rat brains; (**B**) Magnified representative images of GFAP staining in cortex of naïve/sham animals and CCI brains from ipsilateral and contralateral sides of the injury; (**C**) Magnified representative images of GFAP staining in the hippocampus of naïve/sham animals and CCI brains from ipsilateral and contralateral sides of the injury; (**D**) Brain atlas images, showing areas of cortex and hippocampus analysed; (**E**) Quantification of the GFAP+ cell count/ mm^2^ in the cortex of CCI and sham/naïve control rats (2-way ANOVA hemisphere F(1,18) = 67.93 *p* < 0.0001, injury F(1,18) = 104.8 *p* < 0.0001, interaction F(1,18) = 58.42 *p* < 0.0001); (**F**) Quantification of GFAP percentage area covered in the cortex of CCI and sham/naïve control rats (2-way ANOVA hemisphere F(1,19) = 25.57 *p* < 0.0001, injury F(1,19) = 18.95 *p* = 0.0003, interaction F(1,19) = 20.88 *p* = 0.0002); (**G**) Quantification of the GFAP+ astrocytes count/mm^2^ in the hippocampus of CCI and sham/naïve control rat (2-way ANOVA hemisphere F(1,24) = 4.515 *p* = 0.0441, injury F(1,24) = 1.590 *p* = 0.2194, interaction F(1,24) = 1.259 *p* = 0.2730); (**H**) Quantification of GFAP percentage area covered in the hippocampus of CCI and sham/naïve control rats (2-way ANOVA hemisphere F(1,20) = 0.719 *p* = 0.4063, injury F(1,20) = 0.736 *p* = 0.4009, interaction F(1,20) = 0.061 *p* = 0.8071). All data are presented as mean ± SEM, n = 5–8 per group. Two-way ANOVA, **** *p* < 0.0001. Mouse brain atlas images were obtained from the Allen Institute website (www.alleninstitute.org, accessed on 29 September 2022).

**Figure 5 biomolecules-12-01457-f005:**
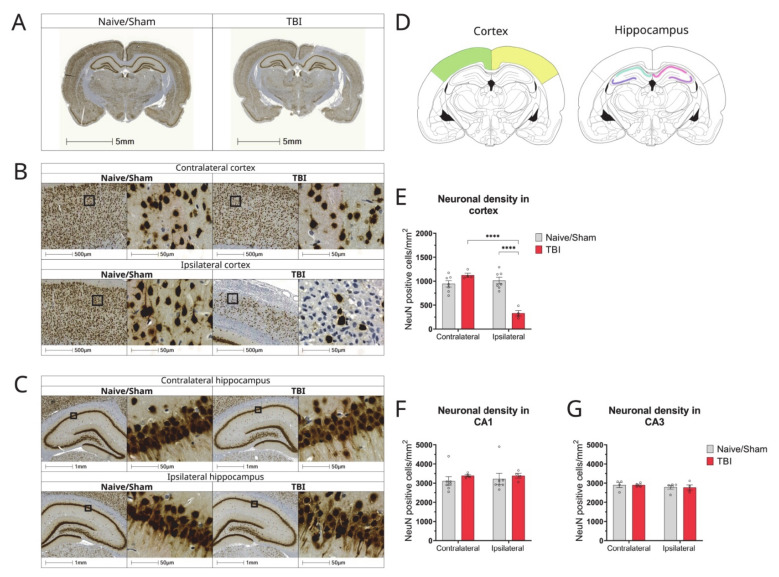
Changes in neuronal density and distribution in cortex and hippocampus of CCI rat brains 2 weeks post-injury. (**A**) Representative whole brain photomicrographs of NeuN+ staining in naïve/control and CCI rat brains; (**B**) Magnified representative images of NeuN staining in cortex of Naïve/sham animals and CCI brains from ipsilateral and contralateral sides of the injury; (**C**) Magnified representative images of NeuN staining in the hippocampus of Naïve/sham animals and CCI brains from ipsilateral and contralateral sides of the injury; (**D**) Brain atlas images, showing areas of cortex and hippocampus analysed; (**E**) Quantification of the NeuN+ cell count/mm^2^ in the cortex of CCI and sham/naïve control rats (2-way ANOVA hemisphere F(1,18) = 27.57 *p <* 0.0001, injury F(1,18) = 13.19 *p* = 0.0019, interaction F(1,18) = 38.87 *p <* 0.0001); (**F**) Quantification of NeuN+ neurons in the CA1 area of the hippocampus of CCI and sham/naïve control rat (two-way ANOVA hemisphere F(1,18) = 0.05058 *p* = 0.8246, injury F(1,18) = 0.7830 *p =* 0.3879, interaction F(1,18) = 0.5585 *p* = 0.8159); (**G**) Quantification of number of neurons in the CA2 area of the hippocampus of CCI and sham/naïve control rats (2-way ANOVA hemisphere F(1,13) = 1.171 *p* = 0.2989, injury F(1,13) = 0.007 *p* = 0.9331, interaction F(1,13) = 0.003 *p* = 0.9520). All data are presented as mean ± SEM, n = 5–7 per group. Two-way ANOVA, **** *p* < 0.0001. Mouse brain atlas images were obtained from the Allen Institute website (www.alleninstitute.org, accessed on 29 September 2022).

**Figure 6 biomolecules-12-01457-f006:**
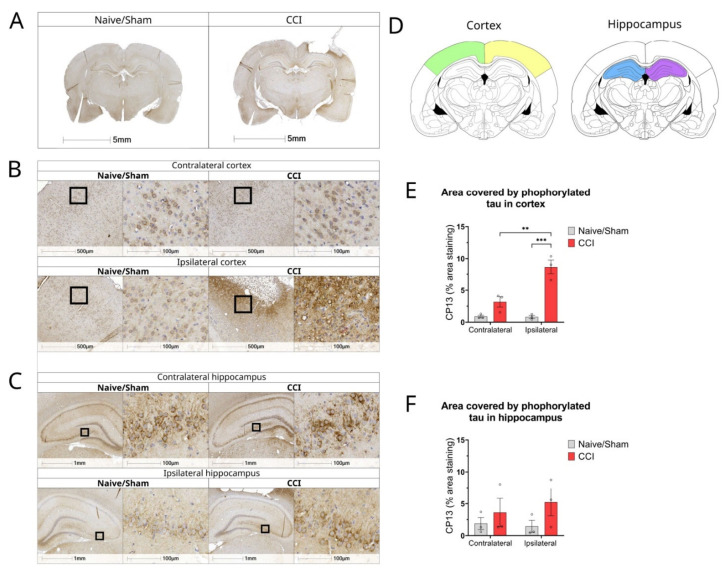
Tau expression is increased in cortex of CCI rat brains 2 weeks post-injury. (**A**) Representative whole brain photomicrographs of CP13 staining in naïve/control and CCI rat brains; (**B**) Magnified representative images of CP13 staining in cortex of naïve/sham animals and CCI brains from ipsilateral and contralateral sides of the injury; (**C**) Magnified representative images of CP13 staining in the hippocampus of naïve/sham animals and CCI brains from ipsilateral and contralateral sides of the injury; (**D**) Brain atlas images, showing areas of cortex and hippocampus analysed; (**E**) Quantification of CP13% area stained in the cortex of CCI and sham/naïve control rats (two-way ANOVA hemisphere F(1,8) = 14.68 *p* = 0.0050, injury F(1,8) = 51.38 *p* < 0.0001, interaction F(1,8) = 15.44 *p* = 0.0044); (**F**) Quantification of CP13% area stained in the hippocampus of CCI and sham/naïve control rat (two-way ANOVA hemisphere F(1,8) = 0.1271 *p* = 0.7306, injury F(1,8) = 2.696 *p* = 0.1393, interaction F(1,8) = 0.3704 *p* = 0.5597);. All data are presented as mean ± SEM, n = 3 per group. Two-way ANOVA, ** *p* < 0.01 *** *p* < 0.001. Mouse brain atlas images were obtained from the Allen Institute website (www.alleninstitute.org, accessed on 29 September 2022).

**Table 1 biomolecules-12-01457-t001:** List of Primers used for mRNA qPCR.

Gene	Forward	Reverse
*BACE1*	GAAGCAGACCCACATTCCGA	CAATGATCATGCTCCCTCCCA
*Tnf*	AGGGATGAGAAGTTCCCAAATG	CACTTGGTGGTTTGCTACGAC
*GAPDH*	CCCCAACACTGAGCATCTCC	GGTATTCGAGAGAAGGGAGGGC
*APP*	GGTGGACTCTGTGCCAGC	TCCGTTCTGCTGCATCTTGG
*Il4*	CAGGGTGCTTCGCAAATTTT	CTCAGTTCACCGAGAACCCC
*Il6*	ATGGATGCTACCAAACTGGAT	TGAAGGACTCTGGCTTTGTCT
*Cx3cl1*	GCAACATCACGTGCCACAAG	GCTGTCTCGTCTCCAGGATGAT
*GSK3* *β*	TCGCCACTCGAGTAGAAGAAA	ACTTTGTGACTCAGGAGA ACT
*MAPT*	AAGAAGCAGGCATCGGAGAC	CCTTGGCTTTCTTCTCGTCA

## Data Availability

The data of this study are available from the corresponding author on reasonable request.

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
