# Peer review of "Traumatic Brain Injury Leads to Alterations in Contusional Cortical miRNAs Involved in Dementia"

_biomolecules, 2022, doi:10.3390/biom12101457_

Round 1

Reviewer 1 Report

The manuscript, “Traumatic brain injury leads to alterations in contusional cortical miRNAs involved in dementia”, looked at the micro RNA expression changes in a controlled cortical impact model of TBI in adult male rats. The group found significant increases in miR-9_1, and reductions in miR-29b_1, miR34a_1, miR-106b_1, miR-107_2, and miR181a_2, some of which are also seen changed in AD. There were also changes in some RNAs that are downstream of the micro RNAs assessed. Finally, the group found microglia and astrocyte activation and reduction in NeuN+ neurons, particularly within the ipsilateral cortex. The group concludes that these changes in micro RNAs are the link between TBI and dementia. These are interesting findings, but there are some concerns.

1)More detail regarding the image analysis would be helpful. How many sections/ animal were analyzed, was the whole region of interest analyzed, if not, how were the areas selected for analysis? It would also be helpful to include a citation for using the HALO software for microglial, astrocyte and neuronal analyses.

2)Including statistical details including the f values, degrees of freedom and p values for data in the results section would be helpful.

3)A discussion point regarding the previous finding indicated that there is a subpopulation of NeuN- neurons in the cortex of Sprague Dawley rats would be helpful. It is unlikely that this subpopulation would alter your findings in the ipsilateral cortex, but it should be noted.

4)A discussion linking the 2w post-injury time point in adult rats to the progression to AD would be helpful as AD typically evolves much later than 2w post-injury.

5)Discussion points regarding the microglia and astrocyte histological data should also be included. How does this data associate with the miRNA findings and findings from other groups.

6) Having additional discussion regarding previous findings in TBI sudies in regard to the miRNAs and their downstream effectors would elevate the discussion. 

Author Response

1)More detail regarding the image analysis would be helpful. How many sections/ animal were analyzed, was the whole region of interest analyzed, if not, how were the areas selected for analysis? It would also be helpful to include a citation for using the HALO software for microglial, astrocyte and neuronal analyses.

At least 3 sections per animal were analysed and more details have been added to the methods section. We had originally included a reference for the HALO software that was used in a previous study from our lab (Donat et al, 2021).

2)Including statistical details including the f values, degrees of freedom and p values for data in the results section would be helpful.

We have now included these statistical details either in the result section or in the figure legends.

3)A discussion point regarding the previous finding indicated that there is a subpopulation of NeuN- neurons in the cortex of Sprague Dawley rats would be helpful. It is unlikely that this subpopulation would alter your findings in the ipsilateral cortex, but it should be noted.

We are not sure what subpopulation of neurons the reviewer is referring to. There are descriptions in the literature of a subpopulation of membrane disrupted neurons following a diffuse TBI using the central fluid percussion injury (CFPI) model (Hernandez et al, Neural Regen Res. 2021 Dec; 16(12): 2409–2410), which is different to our model (CCI being regarded as more focal than diffuse). This effect seems to be resolved (Farkas et al., 2006; Lafrenaye et al., 2014).

4)A discussion linking the 2w post-injury time point in adult rats to the progression to AD would be helpful as AD typically evolves much later than 2w post-injury.

We chose that time point post-injury because it was when the rats started having cognitive impairment. Other potential AD pathological symptoms may appear later, but they are not visible in wild-type rats, because rats do not have amyloid-β deposition, for instance. In fact, in this new version, we have included new data, showing that there is an increase in p-tau in animals 2 weeks post-injury.

5)Discussion points regarding the microglia and astrocyte histological data should also be included. How does this data associate with the miRNA findings and findings from other groups.

We have previously included in the discussion that some miRNAs, such miR-9 have been linked to microglia activation. Another widely studied miRNA in the context of inflammation in microglia and astrocytes is the miR-181 family of miRNAs (Karthikeyan et al, Current Medicinal Chemistry, 2016,), which also appears dysregulated in our study in TBI rats. The changes in miR-9 and miR-181 in TBI from us and other groups are included in the discussion.

6) Having additional discussion regarding previous findings in TBI sudies in regard to the miRNAs and their downstream effectors would elevate the discussion. 

We have now added that miRNAs exhibit promoting or inhibiting effects on the formation of secondary brain damage, such as promotion of neuronal regeneration and apoptosis, alleviation of leakage across the blood-brain barrier (BBB), disruption of intracellular transport, and decreasing the inflammatory response in TBI.

Reviewer 2 Report

In this manuscript, the authors performed a miRNA array in extracts from rats subjected to the controlled cortical impact (CCI) model. Then observed alterations of miRNAs, especially, the expression of miR-9 was significantly upregulated- while miR-29b, miR-34a, miR-106b, 21 miR-181a and miR-107 were downregulated. They also found that the changes in those miRNAs were concomitant with alterations in gene expression of mRNAs of BACE1 or GSK3β, or TNF-α, as well as with an increase in glial activation and neuronal loss in pericontusional areas.

The topic is very interesting and most experiments are properly done. However, I have some doubts on the presentation of the manuscript and on the conclusions reached by the authors based on these experiments. Additional data should be provided to sustain the conclusion and discussion.

1.       The authors did miRNA array and found plenty of miRNAs related to TBI, they also checked the effect of TBI on the density of glial cells and neurons in the brain and characterized neuronal loss in cortex of hippocampus of CCI rats, what’s the correlation between changed miRNAs and the last two parts?

2.       Do the changed miRNAs lead to the changes of glial cells or neurons in density, morphology and distribution in cortex and hippocampus?

3.       Did the changed miRNAs have area specific expression pattern?

4.       It’s better to test protein levels of changed inflammatory markers, neuronal death and genes involved in Aβ generation and tau phosphorylation, such as BACE1 and GSK3. The most important mechanism of miRNAs is to inhibit protein translation.

Author Response

  1. The authors did miRNA array and found plenty of miRNAs related to TBI, they also checked the effect of TBI on the density of glial cells and neurons in the brain and characterized neuronal loss in cortex of hippocampus of CCI rats, what’s the correlation between changed miRNAs and the last two parts? This would be an interesting analysis to do.   Unfortunately, we cannot perform the correlation analysis, since the miRNA extractions and the immunohistochemistry were carried out in different animals
  2. Do the changed miRNAs lead to the changes of glial cells or neurons in density, morphology and distribution in cortex and hippocampus? We cannot prove with our post-mortem approach that the changed miRNAs lead directly to the changes in glial cells or in neuronal density. We have highlighted in the discussion that certain miRNAs have been previously implicated in neuronal survival (for instance miR-29b; Kole and colleagues in 2011). Other miRNAs, such as miR‐9, have been associated with glial activation (Yao et al., 2014). We also detected changes in cytokines and chemokines in our CCI model, but we do not know if the expression of cytokines are responsible for the changes in miRNAs or vice versa. We have carried out some experiments in neuroblastoma cell lines incubated with TNFα and we have not detected differences in the expression of these miRNAs (data do not shown), so we believe the changes in miRNAs are likely responsible for the alterations in glial activation and not the opposite.
  3. Did the changed miRNAs have area specific expression pattern? We only quantified the miRNAs in pericontusional homogenates, so we do not know if the expression in other areas was different. We compared the hemisphere ipsilateral with that contralateral to the injury.
  4. It’s better to test protein levels of changed inflammatory markers, neuronal death and genes involved in Aβ generation and tau phosphorylation, such as BACE1 and GSK3. The most important mechanism of miRNAs is to inhibit protein translation. We have now performed some additional immunohistochemical analysis of BACE1 and phosphorylated tau, in order to see if the protein expression is changed following CCI. We have not observed any changes in the expression of BACE1 2 weeks post-injury (data do not shown). In the discussion, we refer to publications showing changes in BACE1 protein expression immediately after injury, not lasting longer than 3-4 days. However, we did observe increases in phosphorylated tau in the cortex, close to the pericontusional area (new Figure 6).

Reviewer 3 Report

The relationship between TBI and AD may be worth exploring further in future studies involving TBI patients and animals. It should need more detailed reporting of the methods and results.

What is the age and weight of the rats?

How many animals were in each experimental group - naive, sham and CCI? Justify the selected number of animals in the group with statistical power analysis.

Clarify what kind of anaesthesia was used to collect samples from CCI rats.

Specify the perfusion rate.

Specify the concentration of PBS buffer.

Specify the animal food company and supplier.

Update animal welfare information "All studies involving animals were reported in accordance with the ARRIVE guidelines. The experimental procedures were performed in accordance with the guidelines reported in EU Directive 2010/63/EU."

Specify the manufacturer of the DPX mounting solution.

What does control mean - sham or nave?

It would be good to know miRNA changes in sham animals to know the effect of drilling the bone itself on inflammatory markers. Please add information.

What happens to miRNAs and mRNAs in other brain structures, especially the hippocampus? It is known that the hippocampus is affected by AD.

Author Response

What is the age and weight of the rats?

Male Sprague Dawley rats were purchased from Charles River UK at the age of ~ 7 weeks (at ~220-250 g) and housed under standard conditions for ~2 weeks. Animals subjected to surgery/injury had reached 320-375 g.

How many animals were in each experimental group - naive, sham and CCI? Justify the selected number of animals in the group with statistical power analysis.

For the miRNA analysis, we used 4-5 animals per group. For the IHC analysis of glial cells, neurons, BACE1 and tau, we used 5-8 animals per group (for each animal, we analysed 3 brain slices per antibody). The numbers are indicated in the figure legends and were based on previously observed effect sizes. For power calculations we usually use InVivoStat, an R-based statistical package.

Clarify what kind of anaesthesia was used to collect samples from CCI rats.

For tissue extraction, all animals were subjected to deep pentobarbital (Euthanal, Boehringer Ingelheim, UK) anaesthesia. After cessation of all pain reflexes, they were perfused via the transcardial route using ice-cold PBS

Specify the perfusion rate.

For perfusion, we used the Leica Perfusion One system at approximately 50 bar and a 20 g blunt needle.

Specify the concentration of PBS buffer.

We purchased PBS tablets from Sigma, which were 20X concentrated. One tablet dissolved in 200 mL of deionized water yields 0.01 M phosphate buffer, 0.0027 M potassium chloride and 0.137 M sodium chloride, pH 7.4.

Specify the animal food company and supplier.

Rats had ad-libitum access to standard rodent chow (Rat and Mouse No.1 Maintenance, Special diets service, UK).

Update animal welfare information "All studies involving animals were reported in accordance with the ARRIVE guidelines. The experimental procedures were performed in accordance with the guidelines reported in EU Directive 2010/63/EU."

We have updated the welfare information.

Specify the manufacturer of the DPX mounting solution.

Thermo Scientific.

What does control mean - sham or nave? Naïve.

It would be good to know miRNA changes in sham animals to know the effect of drilling the bone itself on inflammatory markers. Please add information.

We cannot provide this information since sham-craniotomy animals were not used specifically due to the pro-inflammatory response. Instead, we are comparing the contralateral side of the injury, which does not have a direct impact, but belongs to the same animal, undergoing the surgery..

What happens to miRNAs and mRNAs in other brain structures, especially the hippocampus? It is known that the hippocampus is affected by AD.

We only extracted miRNAs and mRNAs in the pericontusional areas, which are mostly affected by the injury (see results of IHC, showing no significant changes in hippocampus for certain markers, for instance tau). In the future, we plan experiments using in situ hybridization, allowing us to examine the spatial distribution of those miRNAs and mRNAs. The current manuscript also demonstrates changes in proteins (p-tau in cortex) via IHC in the CCI model.

Reviewer 4 Report

In this article, Authors would investigate the link between Traumatic Brain Injury (TBI) and Alzheimer’s disease (AD), analysing the expression of miRNA related to neurodegenerative disorders (and in particular AD) after injury in TBI animal models. Moreover, the Authors found alterations in gene expression of mRNAs involved in amyloid pathway and tau pathology, and in neuroinflammation.

The manuscript is clear, with a relevance in the field of Neurodegenerative disorders (and in particular of AD) and presented in a well-structured manner. The cited referenced relevant for the study. The experimental design is appropriate to test the hypotheses of the Authors, and the method section is detailed. The figures are appropriate and easy to interpret and understand. The conclusions are consistent with the presented evidences and arguments, and the ethics statements are adequate.

English language is appropriate and understandable, although some corrections in the written form are needed.

Specific comments

16 “hypothesized” instead of “hypothesized”

18 presence of additional space “in extracts of  rats”

21 “,” instead of “-“

22 “we confirmed” instead of “we could confirm”

26 “and in neuroinflammation (TNF-α)” instead of  “neuroinflammation (TNF-α)”

48 “has suggested” instead of “have suggested”

60-61 “and in post-mortem AD samples” instead “and post-mortem”

75 “did not” instead of “do not”

114 please, specify the composition of lysis/binding solution

130 “SYBR green reagent kit” instead of “SRBR green reagent kit”

253 “[…]compared to naïve rats; however, the results did not reach […]”

260 “[…]control brains; however, those changes were not […]”

313 insert a space: “an upregulation” instead “anupregulation”

343 use “[…]using the neuron-343 specific[…]” instead “using a neuron- specific”

361 use  “[…]particularly in the impact region[…]” instead “particularly the impact region”

363 “[…] cortex (p<0.0011, n=7) ipsilateral (p<0.0001, n=4-7) and contralateral […] instead “cortex (p<0.0011, n=7) and ipsilateral (p<0.0001, n=4-7) and contralateral […]”

366 use “[…]regions of TBI animals, AND there was no notable[…]” instead “regions of TBI animals, there was no notable[…]”

383 change “which have been mostly” with “these have been mostly”

421 “miR-132” instead “mir-132”

422 “miR-21” instead “mir-21”

427 “as they are in post- […]” instead “[…], they are in post-[…]”

441 “, so acting as an anti-inflammatory” instead “hence, act as an anti-inflammatory”

444-445 “[…]cell cytoskeleton; therefore, miR-9 increases microtubule instability by targeting stathimin.” instead “[…]cell cytoskeleton, therefore, by targeting stathimin, miR-9 increases microtubule instability[…]”

448 “miR9” instead “mir9”

469 “suggests” instead “suggest”

469-470 “Therefore, these results are key” instead “These results therefore are key[...]”

Author Response

Specific comments

16 “hypothesized” instead of “hypothesized” This has been changed

18 presence of additional space “in extracts of  rats” The space has been cut.

21 “,” instead of “-“ Thanks, that was a typo.

22 “we confirmed” instead of “we could confirm” This has been changed.

26 “and in neuroinflammation (TNF-α)” instead of  “neuroinflammation (TNF-α)” This has been included

48 “has suggested” instead of “have suggested” This has been corrected

60-61 “and in post-mortem AD samples” instead “and post-mortem” This has been corrected

75 “did not” instead of “do not” This has been corrected

114 please, specify the composition of lysis/binding solution. This is supplied with the miRVana kit and there is no information of the composition of this buffer.

130 “SYBR green reagent kit” instead of “SRBR green reagent kit” This typo has been corrected

253 “[…]compared to naïve rats; however, the results did not reach […]” This has been corrected.

260 “[…]control brains; however, those changes were not […]” This has been corrected

313 insert a space: “an upregulation” instead “anupregulation” This has been corrected

343 use “[…] using the neuron-343 specific[…]” instead “using a neuron- specific” This has been corrected

361 use  “[…]particularly in the impact region[…]” instead “particularly the impact region” This has been corrected

363 “[…] cortex (p<0.0011, n=7) ipsilateral (p<0.0001, n=4-7) and contralateral […] instead “cortex (p<0.0011, n=7) and ipsilateral (p<0.0001, n=4-7) and contralateral […]” This has been corrected

366 use “[…]regions of TBI animals, AND there was no notable[…]” instead “regions of TBI animals, there was no notable[…]” . We have rephrased the sentence. Now it reads “NeuN staining in the ipsilateral CA1 and CA3 hippocampal regions of TBI animals was no notable different compared to sham/naïve animals or the contralateral side of the injury”.

383 change “which have been mostly” with “these have been mostly”. Sentence now reads: “these studies have been mostly performed”

421 “miR-132” instead “mir-132” This has been changed

422 “miR-21” instead “mir-21” This has been changed

427 “as they are in post- […]” instead “[…], they are in post-[…]” This has been changed

441 “, so acting as an anti-inflammatory” instead “hence, act as an anti-inflammatory” This has been changed

444-445 “[…]cell cytoskeleton; therefore, miR-9 increases microtubule instability by targeting stathimin.” instead “[…]cell cytoskeleton, therefore, by targeting stathimin, miR-9 increases microtubule instability[…]” This has been changed

448 “miR9” instead “mir9” This has been changed

469 “suggests” instead “suggest” This has been changed

469-470 “Therefore, these results are key” instead “These results therefore are key[...]” This has been changed

Round 2

Reviewer 2 Report

Shahmir Naseer and colleagues did an excellent job addressing the initial concerns raised by the reviewers. The study is improved and clearer after major revision, and merits publication in Biomolecules.

Reviewer 3 Report

Thank you for your answer.